# Characterizing the Relationship between Expression Quantitative Trait Loci (eQTLs), DNA Methylation Quantitative Trait Loci (mQTLs), and Breast Cancer Risk Variants

**DOI:** 10.3390/cancers16112072

**Published:** 2024-05-30

**Authors:** Peh Joo Ho, Alexis Khng, Benita Kiat-Tee Tan, Chiea Chuen Khor, Ern Yu Tan, Geok Hoon Lim, Jian-Min Yuan, Su-Ming Tan, Xuling Chang, Veronique Kiak Mien Tan, Xueling Sim, Rajkumar Dorajoo, Woon-Puay Koh, Mikael Hartman, Jingmei Li

**Affiliations:** 1Genome Institute of Singapore (GIS), Agency for Science, Technology and Research (A*STAR), 60 Biopolis Street, Singapore 138672, Singapore; hopj@gis.a-star.edu.sg (P.J.H.);; 2Department of Surgery, Yong Loo Lin School of Medicine, National University of Singapore, Singapore 119228, Singapore; 3Saw Swee Hock School of Public Health, National University of Singapore and National University Health System, Singapore 117549, Singapore; 4Department of General Surgery, Sengkang General Hospital, Singapore 544886, Singapore; 5Department of Breast Surgery, Singapore General Hospital, Singapore 544886, Singapore; 6Division of Surgery and Surgical Oncology, National Cancer Centre Singapore, Singapore 168583, Singapore; 7Singapore Eye Research Institute, Singapore National Eye Centre, Singapore 169856, Singapore; 8Department of General Surgery, Tan Tock Seng Hospital, Singapore 308433, Singapore; 9Lee Kong Chian School of Medicine, Nanyang Technology University, Singapore 639798, Singapore; 10Institute of Molecular and Cell Biology, Agency for Science, Technology and Research (A*STAR), 61 Biopolis Street, Singapore 138673, Singapore; 11KK Breast Department, KK Women’s and Children’s Hospital, Singapore 229899, Singapore; 12Cancer Epidemiology and Prevention Program, UPMC Hillman Cancer Center, University of Pittsburgh, Pittsburgh, PA 15232, USA; 13Department of Epidemiology, School of Public Health, University of Pittsburgh, Pittsburgh, PA 15261, USA; 14Division of Breast Surgery, Changi General Hospital, Singapore 529889, Singapore; 15Department of Paediatrics, Yong Loo Lin School of Medicine, National University of Singapore, Singapore 119228, Singapore; 16Khoo Teck Puat—National University Children’s Medical Institute, National University Health System, Singapore 119074, Singapore; 17Department of Infectious Diseases, Peter Doherty Institute for Infection and Immunity, University of Melbourne, Melbourne, VIC 3000, Australia; 18Healthy Longevity Translational Research Programme, Yong Loo Lin School of Medicine, National University of Singapore, Singapore 117545, Singapore; 19Singapore Institute for Clinical Sciences, Agency for Science Technology and Research (A*STAR), Singapore 117609, Singapore; 20Department of Surgery, University Surgical Cluster, National University Hospital, Singapore 119074, Singapore

**Keywords:** KCNN4, genetic risk score, methylation, SMR, eQTL, mQTL, breast cancer risk

## Abstract

**Simple Summary:**

In the last few decades, studies have found many common genetic variants linked to breast cancer. Most of these variants are in non-coding regions or regions that do not directly affect gene function. This study aims to see if a polygenic risk score (PRS) based on functional variants could better predict breast cancer risk compared to an established 313-variant breast cancer PRS. Using publicly available summary data from GWASs, and Quantitative Trait Locus studies of gene expression and DNA methylation, we identified 149 variants potentially causally linked to the risk of developing breast cancer; the weighted sum of these variants was designated as the functional variant. The functional PRS was less effective at predicting breast cancer than the established PRS in 3560 breast cancer patients and 3383 non-breast cancer individuals. Combining both scores did not substantially enhance the prediction accuracy of the established PRS.

**Abstract:**

Purpose: To assess the association of a polygenic risk score (PRS) for functional genetic variants with the risk of developing breast cancer. Methods: Summary data-based Mendelian randomization (SMR) and heterogeneity in dependent instruments (HEIDI) were used to identify breast cancer risk variants associated with gene expression and DNA methylation levels. A new SMR-based PRS was computed from the identified variants (functional PRS) and compared to an established 313-variant breast cancer PRS (GWAS PRS). The two scores were evaluated in 3560 breast cancer cases and 3383 non-cancer controls and also in a prospective study (*n* = 10,213) comprising 418 cases. Results: We identified 149 variants showing pleiotropic association with breast cancer risk (eQTL_HEIDI_ > 0.05 = 9, mQTL_HEIDI_ > 0.05 = 165). The discriminatory ability of the functional PRS (AUC_continuous_ [95% CI]: 0.540 [0.526 to 0.553]) was found to be lower than that of the GWAS PRS (AUC_continuous_ [95% CI]: 0.609 [0.596 to 0.622]). Even when utilizing 457 distinct variants from both the functional and GWAS PRS, the combined discriminatory performance remained below that of the GWAS PRS (AUC_continuous_, combined [95% CI]: 0.561 [0.548 to 0.575]). A binary high/low-risk classification based on the 80th centile PRS in controls revealed a 6% increase in cases using the GWAS PRS compared to the functional PRS. The functional PRS identified an additional 12% of high-risk cases but also led to a 13% increase in high-risk classification among controls. Similar findings were observed in the SCHS prospective cohort, where the GWAS PRS outperformed the functional PRS, and the highest-performing PRS, a combined model, did not significantly improve over the GWAS PRS. Conclusions: While this study identified potentially functional variants associated with breast cancer risk, their inclusion did not substantially enhance the predictive accuracy of the GWAS PRS.

## 1. Introduction

In the past decade, genome-wide association studies (GWASs), employing single-nucleotide polymorphism (SNP) arrays and extensive case–control samples, have facilitated the discovery of common low-risk variants associated with breast cancer [1,2,3]. Collaborative entities like the Breast Cancer Association Consortium (BCAC) have pinpointed over 200 SNPs exhibiting significant associations with breast cancer [4]. While the known SNPs account for 18% of the familial relative risk associated with breast cancer, the inclusion of variants reliably imputed from the OncoArray data can substantially increase this proportion to approximately 40% [3,5,6]. The effects of the top breast cancer-associated variants have been aggregated into a genetic metric (i.e., polygenic risk score (PRS)) to evaluate an individual’s predisposition to breast cancer [5].

More research focused on uncovering the specific genetic variants and biological mechanisms responsible for the identified statistical associations with disease risk is needed [7]. A substantial proportion of disease-associated loci are situated in non-coding genome regions [8,9]. While these regions are presumed to influence gene expression regulation, the precise identification of the genes they regulate and the specific cell types or physiological contexts in which this regulation occurs remain elusive [9]. This lack of clarity has impeded the seamless translation of GWAS discoveries into actionable clinical interventions.

Schork et al. demonstrated a consistent enrichment pattern of polygenic effects within various phenotypes, highlighting the highest enrichment for SNPs associated with regulatory and coding genic elements, minimal enrichment in introns, and a negative enrichment for intergenic SNPs [10].

Analytical methods that simulate a functional study in a virtual environment have the potential to identify relevant disease variants. The SMR (summary data-based Mendelian randomization) and HEIDI (heterogeneity in dependent instruments) methodologies can be understood as tests aimed at assessing whether the impact of a SNP on the phenotype is influenced by gene expression or DNA methylation [11,12]. The tests employ summary-level data from GWASs and Quantitative Trait Locus (QTL) studies to assess whether a transcript and a phenotype are linked due to a common causal variant, indicating pleiotropy [11]. SMR has the potential to mitigate confounding and reverse causation, common challenges in traditional association studies. It has demonstrated success in pinpointing gene expression probes or DNA methylation loci that exhibit pleiotropic or potentially causal associations with diverse phenotypes, including neuropathologies associated with corneal thickness, Alzheimer’s disease, and the severity of COVID-19 [13,14,15].

Breast cancer can develop relatively early in life, as compared to other common cancers like prostate and colorectal cancers [16]. National breast cancer screening programs are commonly age-based starting at age 45–50 years [17]. Being able to identify high-risk women using germline variants will empower women to decide on earlier breast cancer screening. In this study, we explore the use of genetic variants that are pleiotropically or potentially causally linked to the risk of developing breast cancer as a polygenic risk score (PRS). The performance of the new PRS will be compared to the established 313-variant breast cancer PRS.

## 2. Results

### 2.1. Pleiotropic Associations—165 DNAm Sites and Eight Genes Associated with Breast Cancer Risk

In the pre-filtering of the SMR analysis, 8556 eQTLs and 126,421 mQTLs had variants in the breast cancer GWAS dataset. Pleiotropic association analyses (i.e., SMR analysis *p* < 5 × 10^−8^ with HEIDI test *p* > 0.05) identified 149 unique variants in eQTLs (*n* = 9; tagging 8 genes) and mQTLs (*n* = 165; tagging 71 genes) (Appendix A, Figure 1). Five of these variants were included in the established 313-SNP breast cancer GWAS PRS.

While SMR analysis did not identify identical variants from the eQTL (Appendix A) and mQTL (Appendix A) datasets, six variants from eQTLs were within a 100 kb window from variants identified in mQTLs (Appendix A, Figure 1), of which two variants from eQTLs tagged the same gene as the variants from mQTLs (Appendix A, Figure 1); (1) *ELL* tagged by rs34010330 (eQTL probe ILMN_1736048 beta_SMR_ = −0.102) and rs1001731353 (closest mQTL probe cg01380346 beta_SMR_ = 0.479), and (2) *KCNN4* tagged by rs62116961 (eQTL probe ILMN_1709937 beta_SMR_ = 0.112), rs112533386 (closest mQTL probe cg15988552 beta_SMR_ = −3.40) (Appendix A, Figure 1).

### 2.2. Larger eQTL Dataset

There are a few eQTL datasets publicly available (accessed on 15 March 2024) [18,19,20]. Here, we found slight differences when we used a larger eQTL blood-based dataset [18]. In the pre-filtering of the SMR analysis, 15,758 eQTLs had variants in the breast cancer GWAS dataset. Similarly, with the CAGE eQTL dataset, SMR analysis using the eQTLGen Consortium data did not identify identical variants from eQTLs and mQTLs. Fifteen variants (tagging 14 genes including *KCNN4*) were identified with pleiotropic association with breast cancer (Appendix A, Appendix A). The SMR-eQTL analysis of *ELL* did not survive the HEIDI test (*p* = 2.19 × 10^−3^) (Appendix A). However, an additional gene, *ATG10*, was identified. This gene was tagged by rs112413913 in SMR-eQTL analysis (beta_SMR_ = −0.097) and rs74906189 in SMR-mQTL analysis (beta_SMR_ = −2.728) (Appendix A). 

### 2.3. The Performance of the Functional PRS Was Inferior to the GWAS PRS

Adding the 149 variants (5 of these 149 variants were in the GWAS PRS) designated as the functional PRS to the GWAS PRS (313 variants) resulted in 457 unique variants in the combined PRS (Appendix A). The discriminatory ability, measured by the area under the receiver operating characteristic curve (AUC), of the functional PRS with all variants and after linkage disequilibrium (LD) pruning was similar (AUC_continuous without LD pruning_ [95% confidence interval (CI)]: 0.540 [0.526 to 0.553] vs. AUC_continuous with LD<0.9_ [95% CI]: 0.541 [0.528 to 0.555]) (Figure 2). No LD pruning was performed for all subsequent analyses.

The discriminatory capability of the functional PRS (AUC_continuous functional PRS_ [95% CI]: 0.540 [0.526 to 0.553]) was inferior to that of the GWAS PRS (AUC_continuous GWAS PRS_ [95% CI]: 0.609 [0.596 to 0.622]), as depicted in Figure 2A. Using the GWAS PRS and the functional PRS as separate variables in a single logistic model did not improve the discrimination of cases and controls (AUC_model_ [95%CI]: 0.609 [0.596 to 0.622]). Utilizing the 457 distinct variants from both the functional PRS and the GWAS PRS, the discriminatory performance remained below that of the GWAS PRS (AUC_continuous, combined_ [95% CI]: 0.561 [0.547 to 0.574]), as illustrated in Figure 2A.

### 2.4. The Discriminatory Ability of the Functional PRS Was Worse Than the GWAS PRS

Fewer variants used resulted in a PRS with worse discriminatory ability. The median AUCs from the 1000 iterations of PRSs (as continuous variable) with 149 (before LD pruning) and 100 (after LD pruning) variants were 0.580 (IQR: 0.574 to 0.585, range: 0.553 to 0.602) and 0.566 (IQR: 0.559 to 0.573), respectively, lower than the GWAS PRS’s (AUC [95%CI]: 0.609 [0.596 to 0.622], range: 0.535 to 0.592) (Figure 3 top panel). 

After accounting for the imbalance in the number of variants, the functional PRS (from eQTLs and mQTLs) continued to perform worse than the simulated GWAS PRS comprising only 149 variants (Figure 3 top panel). Similar results were observed when we studied the discriminatory ability of using the PRS to classify women into a binary group of high/low risk, where women with a PRS score above the 80th centile of the control group were classified as high risk (Figure 3 bottom panel).

### 2.5. Performance of PRS Was Not Dependent on Weights Used; However, Different Individuals Were Identified as High Risk

When the same 313 variants were used, the discriminatory ability of the GWAS PRS was not dependent on the weights used. Limiting the number of high-risk individuals from the control group to 20%, the discriminatory ability of the GWAS PRS with the same variants but different weights was the same to the third decimal place, 0.560 [0.550 to 0.570], for the AUC of the GWAS PRS (as continuous variable) with weights from Mavaddat et al. [5] and weights from Michailidou et al. [3], respectively (Figure 2, Table 1). However, the individuals identified as high risk were different by 5% and 15% for cases and controls, respectively (Appendix A). Considering the PRS as a continuous variable resulted in higher AUCs but still did not differentiate the two differently weighted GWAS PRSs.

### 2.6. Individuals Identified as High Risk by PRS

We classified individuals as high risk if their PRS exceeded the 80th centile value of the PRS distribution in the controls (i.e., a binary variable high/low risk was created). The GWAS PRS with 313 variants identifies 8% more cases as high risk than the functional PRS with 149 variants (Figure 4A). The functional PRS can identify a separate group of high-risk cases (12%) from the GWAS PRS (Figure 4A). However, there was a corresponding larger increase in the proportion of controls (13%) that were identified as high risk (Figure 4A). 

In the analysis, where the number of high-risk individuals in the control population was fixed at 20%, the number of cases identified as high risk by the combined PRS with 457 variants (i.e., from both functional PRS and GWAS PRS) were fewer than the GWAS PRS (n_combined_ = 933 vs. n_GWAS PRS_ = 1139). Notably, 26% of the exact cases identified by the combined PRS and the GWAS PRS were not the same (Figure 4B, Table 1).

The PRS sensitivity and specificity for the functional, GWAS, and combined PRS without LD pruning are as follows.

At the most optimal threshold for discrimination between cases and controls (by Youden’s J statistics), sensitivity was the lowest for the combined PRS (sensitivity_continuous,LD<0.9_ = 0.522) and the highest for the functional PRS (sensitivity_continuous_ = 0.651) in the case–control dataset (Table 1). Specificity was the highest for the functional PRS (specificity_continuous,LD<0.9_ = 0.680) and the lowest for the GWAS PRS (specificity_continuous_ = 0.428) (Table 1). 

In the case where we selected individuals with a PRS score above that of the 80th centile of the controls (i.e., specificity_binary_ = 0.8), sensitivity was the lowest for the functional PRS (sensitivity_binary_ = 0.232) and the highest for the GWAS PRS (sensitivity_binary_ = 0.320) (Table 1).

### 2.7. The Validation of the Findings from the Case–Control Study in a Prospective Cohort

The number of variants available in the SCHS dataset differs from that of SGBCC (Appendix A). Of the 149 variants in the SMR-based PRS, 146 variants were in the SCHS dataset, of which 144 were in SGBCC. Of the 313 variants in the established PRS, 290 variants were in the SCHS dataset, of which 285 were in SGBCC.

We observed similar findings in the SCHS prospective cohort. The GWAS PRS (PRS313) outperformed the functional PRS (AUC_GWAS PRS_: 0.595 [0.567 to 0.623] vs. AUC_functional PRS_: 0.568 [0.541 to 0.596]) (Table 1). However, the highest-attaining PRS was the combined PRS with 431 variants (AUC_combined_: 0.603 [0.575 to 0.630]) (Table 1). Notably, the confidence interval of AUC_combined_ includes the estimate of AUC_GWAS PRS_; the improvement is unlikely to be significant. 

In SCHS, the prospective cohort of 10,213, with 418 women who developed breast cancer, sensitivity was similar for the functional PRS (sensitivity_continuous_ = 0.688), GWAS PRS (sensitivity_continuous_ = 0.655), and combined PRS (sensitivity = 0.648) (Table 1). In addition, the specificity was the highest for the combined PRS (specificity_continuous_ = 0.531) followed by the GWAS PRS (specificity_continuous_ = 0.490) (Table 1). 

### 2.8. The Discriminatory Ability of the PRS Derived from Blood, Adipose, and Breast Tissues of the 46 Variants in Our Case–Control Dataset

Seventeen variants from either the GWAS PRS (15 variants), SMR-eQTL (two variants: chr6:130349119 [tagging *L3MBTL3*] and chr15:91531995 [tagging *RCCD1*]), or SMQ-mQTL (1 variant: chr12:14413931) were in the 46 variants identified by Ferreira et al. [21] (Appendix A). The PRS with only 46 variants from Ferreira has a discriminatory ability of 0.557 [0.544 to 0.571] (Appendix A). The addition of the 46 variants to the functional PRS, GWAS PRS, or both did not improve the discriminatory ability of the GWAS PRS alone (Appendix A). 

Of the four genes identified by Ferreira et al. specifically in breast tissues, *RCCD1* was identified in both SMR-eQTL analysis using the CAGE dataset and eQTLGen Consortium data [21]. *ATG10* was identified in the eQTLGen Consortium data. 

## 3. Discussion

Genome-wide association studies (GWASs) have revealed a plethora of genetic variations influencing susceptibility to diverse human diseases and traits [22]. These effects of these variants have been aggregated into PRSs, which are powerful tools for predicting risk [4]. Not all GWAS-derived variants are necessarily functional; many occur in non-coding regions of the genome or in regions that do not directly affect gene function [23,24]. We performed post-GWAS annotations to select the most likely functional variants to compute an alternative functional PRS for breast cancer and compared it to the established 313-variant breast cancer GWAS PRS.

Several statistical methods have been suggested to prioritize signals from GWASs by integrating various functional evidence [23]. This approach aims to prioritize variants with modest effect sizes but exhibiting functional characteristics, distinguishing them from variants with comparable effect sizes but a lower likelihood of being functional. Our SMR analysis identified 149 unique variants associated with breast cancer risk in both eQTL and mQTL datasets, suggesting potential functional relevance. However, the functional PRS, despite incorporating variants associated with functional elements, demonstrated inferior discriminatory performance compared to the GWAS PRS, even when combining both types of variants. This trend was similarly seen in an independent prospective cohort.

To examine how the discriminatory ability of PRSs fluctuates with changes in the variant composition and weightings of PRSs, we constructed a PRS using subsets of variants, varying numbers of variants, and different weights. The discriminatory ability decreased as the number of variants in the PRS decreased, with simulations using fewer variants showing lower median AUC values compared to the original GWAS PRS. Even after accounting for the number of variants, the functional PRS consistently underperformed the GWAS PRS, all subsets of 149 variants from the GWAS PRS outperforming the functional PRS. We also found that the discriminatory ability of the PRS, when using the same variants, was not substantially influenced by the weights applied. Despite similar discriminative abilities, the exact individuals identified as high risk differed by five percent.

Our approach to classifying individuals as high risk based on the 80th centile value of the PRS distribution in controls yields noteworthy insights into the performance of different PRS models. The GWAS PRS, encompassing 313 variants, identifies a higher proportion of breast cancer cases as high risk compared to the functional PRS with 149 variants. This underscores the GWAS PRS’s ability to effectively capture individuals predisposed to breast cancer. Conversely, the functional PRS, while identifying a separate group of high-risk cases, is associated with a notable increase in the proportion of controls flagged as high risk. While a higher sensitivity in detecting cases is advantageous, the higher number of flagged controls may result in a greater number of false positives, potentially leading to unnecessary interventions, increased healthcare costs, and heightened psychological distress for individuals incorrectly identified as high risk [25,26,27]. The implications highlight the challenge of achieving a balanced risk prediction model that optimizes sensitivity for case identification while minimizing the false positive rate in controls [28]. Careful consideration is needed to strike the right balance between sensitivity and specificity in order to avoid unnecessary interventions and optimize the predictive accuracy of the PRS model.

Our study employs a comprehensive approach by integrating genetic and functional data, including eQTL and mQTL datasets, to assess breast cancer risk. Conducting pleiotropic analysis helps identify variants associated with both gene expression and methylation, providing a more nuanced understanding of potential functional relevance. Large datasets with significant sample sizes are used, enhancing statistical power and the generalizability of the findings. The validation of the findings in a prospective cohort (SCHS) adds robustness to the study, supporting the generalizability of the results beyond the initial case–control dataset. This study rigorously evaluates the discriminatory performance of different PRS models, including sensitivity, specificity, and AUC, providing a comprehensive understanding of their predictive capabilities.

This study’s findings may be specific to the population under investigation, limiting generalizability to other populations with different genetic backgrounds and environmental exposures. Of note, the case–control and prospective cohort analyses were performed on populations of Asian descent, while the SMR analyses were performed on predominantly European datasets. Nonetheless, the European-derived breast cancer GWAS PRS has been shown to be applicable for Asian populations [29,30]. Assigning functional significance to identified variants can be challenging, and there may be limitations in fully characterizing the biological impact of specific genetic markers. This study relies on available functional genomics data, and the absence of certain functional annotations or complete information on all potential functional elements may limit the interpretation of results. In addition, there is a paucity of expression and methylation data in actual breast tissue. The variants derived from the eQTLs and mQTLs of tissue samples could improve the performance of the functional PRS. Furthermore, normal breast tissue is highly heterogeneous, including epithelial cells of several lineages (e.g., lobular, basal, luminal, stem cells) and numerous stromal cell types [31]. Single-cell expression and methylation studies could uncover variants that are more directly relevant to breast carcinogenesis. At the current state of the art, improving upon the GWAS-derived PRS using largely blood-derived eQTL and mQTL data appears difficult. 

## 4. Materials and Methods

### 4.1. Summary Data-Based Mendelian Randomization (SMR)

The breast cancer risk GWAS comprised 122,977 cases and 105,974 controls of European descent [3]. Summary statistics were available for 11,792,542 variants, where 20,933 were significant at *p* < 5 × 10^−10^. 

For QTL datasets, we used the lite version of the CAGE eQTL summary data (*n* = 2765, 36,754 SNPs with *p* < 1 × 10^−5^ are included) [29] and the whole blood mQTL dataset (*n* = 1175, 126,457 SNPs with mQTL *p*-values < 1 × 10^−10^) used in Hannon et al. [32]. We also used the largest eQTL database QTLGen Consortium data (as of March 2024), which pools data from 37 sources, totaling 25,482 whole blood and 6202 peripheral blood mononuclear cell samples. The dataset includes 8,932,843 variants and 19,250 genes [18].

The integration of information from GWASs, eQTLs, and mQTLs follows a sequential procedure. We first identified associations between gene expressions and breast cancer risk by applying SMR on cis-eQTL and GWAS datasets, respectively. Only results that passed (p_HEIDI_ > 0.05) the test for heterogeneity in dependent instruments (HEIDI) were considered pleiotropic associations and not mere linkage [12]. Concurrently, we identified pleiotropic associations between DNAm sites and breast cancer, using mQTL and GWAS datasets, respectively. 

Following that, we evaluated whether the variants identified from eQTLs and mQTLs were identical. Considering that variants identified through summary statistics may differ, we examined the linkage disequilibrium between the variants designated as significant results (pSMR < 5 × 10^−8^) in the SMR analysis. The linkage disequilibrium (LD) between variants was determined using the R package “LDlinkR” [33].

The visualization of regions with significant associations between gene expression and traits and DNA methylation and traits was performed using the R package “karyoploteR” [34]. Annotation was conducted using the R package “TxDb.Hsapiens.UCSC.hg19.knownGene” Carlson M, Maintainer BP (2015). TxDb.Hsapiens.UCSC.hg19.knownGene: Annotation package for TxDb object(s). R package version 3.2.2. TxDb.Hsapiens.UCSC.hg19.knownGene: Annotation package for TxDb object(s).

### 4.2. Selection of Functional Variants for Breast Cancer PRS Construction

Blood-based variants that passed the SMR test (*p* < 5 × 10^−8^) and HEIDI test (*p* > 0.05) for either the eQTL or mQTL analysis were included in our SMR-based variants for breast cancer risk PRS construction. To account for dependency between variants selected from two independent sources, we tested the PRS with a subset of variants with linkage disequilibrium (LD) < 0.9. 

Gene expression changes due to disease are more accurately studied in the respective tissue the disease occurs in—here, eQTLs from breast tissues would be preferred. Ferreira et al. identified 46 variants associated with breast cancer risk, using a joint association analysis [21], of which four sentinel risk variants were in LD (>0.8) with sentinel eQTLs in breast tissue for *ATG10*, *PIDD1*, *RCCD1*, and *APOBEC3B* [21]. Here, we assessed the discriminatory ability of the PRS of the 46 variants in our case–control dataset. 

### 4.3. Polygenic Risk Score (PRS)

The *PRS* is estimated as the weighted sum of effect alleles in *n* single-nucleotide polymorphisms (SNPs) found to be associated with breast cancer (Equation (1)).
(1)$$PRS=\beta_{1}x_{1}+\beta_{2}x_{2}+\cdots +\beta_{k}x_{k}+\cdots +\beta_{n}x_{n}$$
where *x_k_* is the dosage of the risk allele (0–2) for SNP *k*, and *β_k_* is the corresponding weight. The weights of the *n* SNPs for overall breast cancer risk were obtained from the GWAS results published by Michailidou et al. [3]. PRSs were calculated using plink1.9 with the scoresum option [35].

### 4.4. The Performance of the Functional PRS in a Case–Control Study

Breast cancer patients were sourced from the Singapore Breast Cancer Cohort (SGBCC) [36]. Controls, who were free from breast cancer at the time of enrollment into the Singapore Multi-Ethnic Cohort Phase 2 (MEC) study, were matched based on sex, ethnicity, and age density to the cases [37].

### 4.5. Cases—Singapore Breast Cancer Cohort (SGBCC)

Between 2010 and 2016, 7768 female breast cancer patients were recruited from six restructured hospitals: National University Hospital, KK Women’s and Children’s Hospital, Tan Tock Seng hospital, Singapore General Hospital, National Cancer Centre Singapore, and Changi General Hospital. The cohort is described in [36]. The majority (76%) of breast cancer cases in Singapore are treated in these hospitals [36]. Seventy-six per cent (*n* = 5931) of these women provided blood or saliva samples. We excluded 1441 individuals who were not genotyped (biospecimens not available for retrieval from repository, or biospecimens were not collected before genotyping or sequencing experiments), 10 who were genetic duplicates, 6 who failed genotyping quality control, 5 who were duplicated due to recruitment at multiple hospitals, 20 who withdrew consent, and 889 not genotyped on Illumina OncoArray-500 K Beadchip. A total of 3560 breast cancer patients from SGBCC were included in this study.

### 4.6. Controls—Singapore Multi-Ethnic Cohort Phase 2 (MEC2) Study

The MEC2 cohort is described in [37]. In brief, the MEC2 study is a population-based cohort of Chinese, Malay, and Indian adults aged between 21 and 75 years. The cohort is designed to be representative of the general adult population of Singapore. The controls used in this study were ethnicity and age (+/− 5 years) matched to SGBCC breast cancer cases [37]. Among 4099 women who were never diagnosed with breast cancer, 3383 were included in our study after excluding genetic duplicates (*n* = 1), males (*n* = 9), and women not genotyped on Illumina OncoArray-500 K Beadchip (*n* = 709). 

All individuals had genetic information that met quality control standards, as illustrated in Appendix A. Here, we included only the individuals who were genotyped on the Illumina OncoArray-500 K Beadchip (3560 cases and 3383 controls).

### 4.7. DNA Isolation and Genotyping

For buffy coat samples (peripheral blood samples) from SGBCC and MEC2, DNA isolation was performed according to manufacturer’s instructions using FlexiGene DNA kit, Qiagen, or Promega’s Maxwell 16 Blood DNA Purification Kit. For the saliva samples from SGBCC, DNA isolation was performed using Oragene and prepIT•L2P (DNA Genotek). For the case–control study, genotyping was performed on the Illumina OncoArray-500 K Beadchip, as per previously described [3]. The imputation of ~21 M variants was conducted using the 1000 Genomes Project (Phase 3) reference panel [3]. Variants (~11.8 M) retained for analysis had a minor allele frequency (MAF) > 0.5% and an imputation quality score > 0.3 [3]. 

### 4.8. Performance Assessment of PRS

We assessed the discriminatory performance of the breast cancer PRS in our case–control dataset by examining the area under the receiver operating characteristic curve (AUC). Three PRSs were evaluated: (1) the new SMR-based breast cancer PRS (i.e., functional PRS), (2) the established 313-SNP breast cancer PRS (i.e., GWAS PRS), and (3) a union of the variants in the two PRSs. We repeated the analysis after pruning away variants in high linkage disequilibrium for the newly developed functional PRS (>0.9). 

The weights for the original breast cancer GWAS PRS described in Mavaddat et al. were developed using lasso regression (i.e., not directly from the GWAS summary statistics). To examine the dependency of the discriminatory performance of the PRS on the weights, we compared a PRS calculated using the published breast cancer weights in Mavaddat et al. [5] and a GWAS PRS weighted by GWAS summary results in Michailidou et al. (used in other analyses here) [5].

The discriminatory abilities of PRSs were assessed as continuous variables (in standard deviation) and discrete variables identifying high-risk individuals. For each PRS, using only the scores of controls (women without breast cancer), we identified the score corresponding to the 80th centile. Cases and controls whose score exceeds this threshold were considered individuals at high risk. 

Our selection of the threshold (80th centile) to classify individuals as high risk is not the most optimal to maximize the performance of PRSs. For the continuous form of PRSs, we reported the sensitivity and specificity for each PRS based on Youden’s J statistics [38].

### 4.9. Simulation Study to Address Imbalance in Number of Variants in Each PRS

To address the imbalance in the number of variants included in the SMR-based functional and established 313-SNP GWAS PRS, we performed simulations where only an *i* number of variants were randomly included from the PRS with more variants for the score computation, where *i* is the number of variants in the PRS with less variants. A total of 1000 iterations were performed for *i* variants selected without replacement. 

### 4.10. The Validation of the Findings from the Case–Control Study in a Prospective Cohort

We validated the results from the case–control study in the Singapore Chinese Health Study (SCHS) [39]. The SCHS cohort is a prospective cohort, with 63,257 Chinese participants recruited between 1993 and 1998. The participants were mainly from Southern Han Chinese of Hokkien and Cantonese dialect groups in Singapore (https://sph.nus.edu.sg/research/cohort-schs/, accessed on 22 April 2024). For our validation, we selected from 35,298 (56%) females, aged between 45 and 74 years at enrollment. Among these women, 11,280 had genotyping information available [40]. To obtain information about breast cancer occurrence, linkage with the Singapore Cancer Registry was performed. A further exclusion of participants with a history of any cancer at enrollment resulted in 10,213 available for this analysis. As of 31 December 2015, a total of 418 breast cancer cases were diagnosed. 

DNAs from peripheral blood in SCHS samples were extracted using QIAamp DNA Blood kits (Qiagen, Valencia, CA, USA). Genotyping information was generated using the Illumina Global Screening Array [40]. The imputation of additional autosomal variants was conducted with IMPUTE v2 using the cosmopolitan 1000 Genomes reference panel (Phase 3). PRSs were calculated in R for the SCHS prospective study.

All analyses were conducted using the R package version 3.2.2.

## 5. Conclusions

In summary, our results suggest that the GWAS PRS provides better discriminatory performance for breast cancer risk prediction compared to the SMR-derived functional PRS. Features that are not known to be or are not directly related to gene expression and DNA methylation may be included in the agnostic PRS (GWAS PRS) but not the functional PRS resulting in the agnostic PRS outperforming the functional PRS in discrimination. While this study identified potentially functional variants associated with breast cancer risk, their inclusion did not enhance the predictive accuracy of the GWAS PRS. Future work may need to further explore and refine the incorporation of functional information into PRS models for more accurate breast cancer risk prediction.

## Figures and Tables

**Figure 1 cancers-16-02072-f001:**
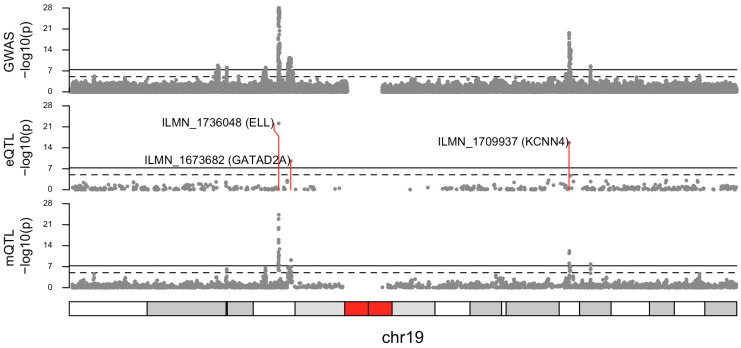
Plot of −log10 *p*-values from breast cancer risk genome-wide association study (GWAS) (**top**), summary data-based Mendelian randomization (SMR) analysis for associations between gene expression and breast cancer risk (**middle**), and SMR analysis for associations between DNA methylation and breast cancer risk (**bottom**). Only variants from the SMR analysis which passed the HEIDI test (pHEIDI > 0.05) are shown.

**Figure 2 cancers-16-02072-f002:**
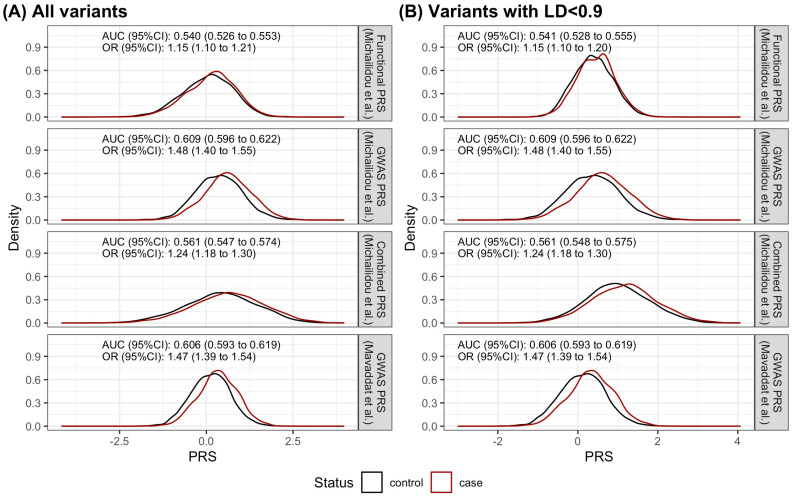
The distribution of breast cancer risk polygenic risk scores (PRSs), presented in a density plot, by case–control status. (1) Functional PRS: variants are significant in either the summary data-based Mendelian randomization analysis of eQTLs or mQTLs, (2) the GWAS PRS weighted by Michailidou et al. [3]: available variants from the published breast cancer risk PRS, (3) Combined PRS: the combination of (1) and (2) using weights from Michailidou et al. [3], and (4) the GWAS PRS weighted by Mavaddat et al. [5]: available variants from the published breast cancer risk PRS. (**A**) All variants from eQTLs and mQTLs. (**B**) Variants with linkage disequilibrium (LD) < 0.9. AUC: area under the receiver operating characteristic curve, CI: confidence interval, OR: odds ratio. The area under the density curve equals one.

**Figure 3 cancers-16-02072-f003:**
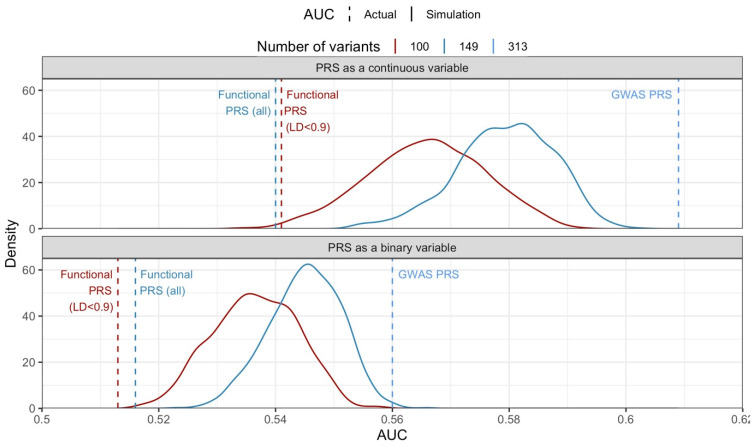
The distributions of AUCs from simulations of 1000 iterations are presented as density plots, by the number of variants used (100 variants in red or 149 variants in blue). In each iteration, a random sample of 100 variants (red solid line) or 149 variants (blue solid line) were used to calculate the polygenic risk score (PRS) in the case–control study, and the AUC was calculated based on this PRS. We sampled variants without replacement from the 313 variants in the GWAS PRS. All PRSs were calculated with weights from Michailidou et al. [3] AUCs using the functional PRSs and the GWAS PRS are indicated by the dashed vertical lines. AUC: area under the receiver operating characteristic curve, LD: linkage disequilibrium. The area under the density curve equals one.

**Figure 4 cancers-16-02072-f004:**
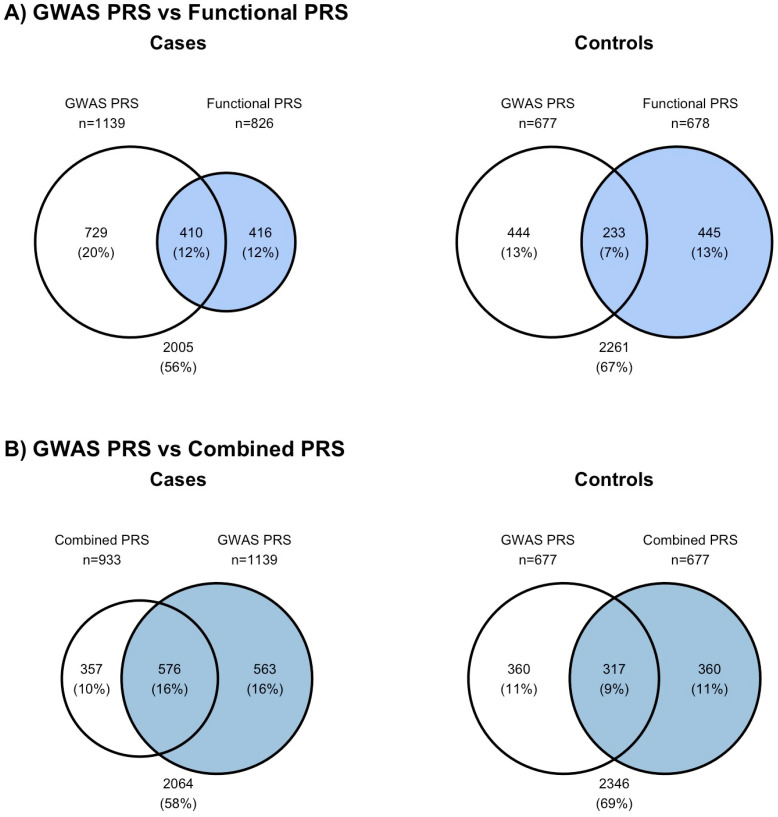
Individuals identified as at a high risk of breast cancer. High risk was based on the threshold at the 80th centile of the control group for each score (i.e., the top 20% [*n* = 667] of the controls for each score are high-risk individuals). (**A**) identifies overlaps between the high-risk individuals identified by the GWAS PRS and the functional PRS. (**B**) identifies overlaps between the high-risk individuals identified by the GWAS PRS [306 variants] and the combined PRS. All PRSs were calculated with weights from Michailidou et al. [3].

**Table 1 cancers-16-02072-t001:** The performance of polygenic risk scores (PRSs) measured by the area under the receiver operating characteristic curve (AUC) and the odds ratio (OR). For the continuous PRS, the OR was calculated per standard deviation change. *n*: number of variants, CI: confidence interval. ^ The split for the binary PRS was based on the 80th centile of the PRS distribution in individuals without breast cancer. * Corresponding to the threshold selected by the Youden’s J statistics, when the PRS is a continuous variable.

		Case–Control Study					Prospective Cohort Study				
PRS	Weights	*n*	AUC (95%CI)	OR (95%CI)	Sensitivity *	Specificity *	*n*	AUC (95%CI)	OR (95%CI)	Sensitivity *	Specificity *
Continuous											
GWAS PRS	Mavaddat et al. [5]	313	0.606 (0.593 to 0.619)	1.47 (1.39 to 1.54)	0.688	0.469	290	0.592 (0.564 to 0.621)	1.42 (1.29 to 1.57)	0.324	0.820
Functional PRS	Michailidou et al. [3]	149	0.540 (0.526 to 0.553)	1.15 (1.10 to 1.21)	0.651	0.417	146	0.568 (0.541 to 0.596)	1.28 (1.16 to 1.41)	0.688	0.422
GWAS PRS	Michailidou et al.	313	0.609 (0.596 to 0.622)	1.48 (1.40 to 1.55)	0.578	0.581	290	0.595 (0.567 to 0.623)	1.43 (1.29 to 1.57)	0.655	0.490
Combined PRS	Michailidou et al.	457	0.561 (0.547 to 0.574)	1.24 (1.18 to 1.30)	0.522	0.572	431	0.603 (0.575 to 0.630)	1.44 (1.30 to 1.59)	0.648	0.531
Functional PRS, LD < 0.9	Michailidou et al.	100	0.541 (0.528 to 0.555)	1.15 (1.10 to 1.20)	0.390	0.680	98	0.564 (0.537 to 0.592)	1.25 (1.13 to 1.38)	0.643	0.485
Combined PRS, LD < 0.9	Michailidou et al.	401	0.561 (0.548 to 0.575)	1.24 (1.18 to 1.30)	0.489	0.615	376	0.597 (0.569 to 0.624)	1.42 (1.29 to 1.57)	0.690	0.471
Binary ^											
GWAS PRS	Mavaddat et al.	313	0.560 (0.550 to 0.570)	1.88 (1.69 to 2.10)	0.320	0.8	290	0.565 (0.543 to 0.588)	1.98 (1.60 to 2.44)	0.331	0.8
Functional PRS	Michailidou et al.	149	0.516 (0.506 to 0.525)	1.21 (1.07 to 1.35)	0.232	0.8	146	0.538 (0.516 to 0.560)	1.53 (1.23 to 1.90)	0.276	0.8
GWAS PRS	Michailidou et al.	313	0.560 (0.550 to 0.570)	1.88 (1.68 to 2.10)	0.320	0.8	290	0.558 (0.536 to 0.581)	1.85 (1.50 to 2.29)	0.317	0.8
Combined PRS	Michailidou et al.	457	0.531 (0.521 to 0.541)	1.42 (1.27 to 1.59)	0.262	0.8	431	0.551 (0.529 to 0.574)	1.73 (1.40 to 2.15)	0.302	0.8
Functional PRS, LD < 0.9	Michailidou et al.	100	0.513 (0.504 to 0.523)	1.17 (1.04 to 1.31)	0.227	0.8	98	0.531 (0.510 to 0.552)	1.42 (1.14 to 1.77)	0.262	0.8
Combined PRS, LD < 0.9	Michailidou et al.	401	0.531 (0.522 to 0.541)	1.43 (1.27 to 1.60)	0.263	0.8	376	0.550 (0.528 to 0.572)	1.71 (1.38 to 2.12)	0.300	0.8

## Data Availability

CAGE eQTL summary data (*n* = 2765) (10.1016/j.ajhg.2016.12.008): https://yanglab.westlake.edu.cn/data/SMR/cage_eqtl_data_lite_hg19.tar.gz, accessed on 1 December 2023. Whole blood mQTL dataset used in Hannon et al. (10.1016/j.ajhg.2018.09.007): https://yanglab.westlake.edu.cn/data/SMR/US_mQTLS_SMR_format.zip, accessed on 1 December 2023. GWAS Summary Results: Breast Cancer Risk (2017) in Europeans: https://bcac.ccge.medschl.cam.ac.uk/bcacdata/oncoarray/oncoarray-and-combined-summary-result/gwas-summary-results-breast-cancer-risk-2017/, accessed on 1 December 2023. eQTLGen Consortium data (2021) (10.1038/s41588-021-00913-z) https://eqtlgen.org/cis-eqtls.html. The data that support the findings of our study are available from the corresponding authors of the study upon reasonable request (Dr Jingmei Li, lijm1@gis.a-star.edu.sg).

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
