# Peer review of "Characterizing the Relationship between Expression Quantitative Trait Loci (eQTLs), DNA Methylation Quantitative Trait Loci (mQTLs), and Breast Cancer Risk Variants"

_cancers, 2024, doi:10.3390/cancers16112072_

Round 1

Reviewer 1 Report

Comments and Suggestions for Authors

This manuscript supports the conclusion that functional PRS derived from eQTL and mQTL studies do not necessarily have improved performance over GWAS-derived PRS. While this could be seen as a "negative" finding, it is of potential interest to cancer epidemiologists and geneticists.

One important limitation of the study, which should be addressed more explicitly in a revised manuscript, is that most eQTL and mQTL data are obtained from studies of blood specimens. There is a paucity of expression and methylation data in actual breast tissue - which could improve the performance of functional PRS. Furthermore, normal breast tissue is highly heterogeneous, including epithelial cells of several lineages (e.g. lobular, basal, luminal, stem cells) and numerous stromal cell types. Perhaps future single cell expression and methylation studies could uncover variants that are more directly relevant to breast carcinogenesis. At the current state of the art, improving upon GWAS-derived PRS using largely blood-derived eQTL and mQTL data appears to be difficult.

Author Response

We thank the reviewer for highlighting an important limitation that readers will appreciate. We have attached the tracked changes version of the manuscript.

We have added the limitation at the end of the last second paragraph: “In addition, there is a paucity of expression and methylation data in actual breast tissue. The variants derived from eQTL and mQTL of tissue samples could improve the performance of functional PRS. Furthermore, normal breast tissue is highly heterogeneous, including epithelial cells of several lineages (e.g. lobular, basal, luminal, stemcells) and numerous stromal cell types (10.1186/bcr2755). Single-cell expression and methylation studies could uncover variants that are more directly relevant to breast carcinogenesis. At the current state of the art, improving upon GWAS-derived PRS using largely blood-derived eQTL and mQTL data appears difficult.”

Reviewer 2 Report

Comments and Suggestions for Authors

This manuscript explores the functional genetic variants associated with the risk of developing breast cancer through the lens of the polygenic risk score (PRS). The work presented here is meticulously prepared and comprehensive, though I do have a few minor suggestions for further enhancement.

  1. Abstract: While the Abstract appears to have subsections, the first subsection under "Purpose" is absent.
  2. Introduction, L80: "Schork and colleagues" should be revised to "Schork et al." for consistency.
  3. Introduction: The authors should provide justification for why they specifically chose to evaluate PRS with breast cancer rather than other types of cancer.
  4. Table 1 contains some typographical errors; thorough editing is advised for the revision.
  5. Larger eQTL dataset, L134-135: The URLs provided should be moved to the reference section.
  6. Figures 2 and 3: It would be helpful to include the unit for "Density" on the y-axis. Additionally, ensure consistency in the y-axis scales between the two figures; one may represent relative density while the other should display percentage density.
  7. Figure 4: The proportion of the area does not accurately reflect the number of cases; consider adjusting this for improved clarity.
  8. Polygenic risk score (PRS), L389: Consider labeling the equation for clarity and reference.

Comments on the Quality of English Language

No comment.

Author Response

We thank the reviewer for the kind assessment. We have attached the tracked changes version of the manuscript. The following is our point-by-point response. 

  1. Abstract: While the Abstract appears to have subsections, the first subsection under "Purpose" is absent.
    Response: We have added the subsection “Purpose:” to the abstract.

  2. Introduction, L80: "Schork and colleagues" should be revised to "Schork et al." for consistency.
    Response: We have revised "Schork and colleagues" to "Schork et al."

  3. Introduction: The authors should provide justification for why they specifically chose to evaluate PRS with breast cancer rather than other types of cancer.
    Response: We have added a justification to the last paragraph of the introduction: “Breast cancer can develop relatively early in life, as compared to other common cancers like prostate and colorectal cancers (10.1002/ijc.34671). National breast cancer screening programs are commonly age-based starting at age 45-50 years (10.3322/caac.21660). Being able to identify highrisk women using germline variants will empower women to decide on earlier breast cancer screening. In this study, we explore the use of genetic variants that are pleiotropically or potentially causally linked to the risk of developing breast cancer as a polygenic risk score (PRS). The performance of the new PRS will be compared to the established 313-variant breast cancer PRS.”

  4. Table 1 contains some typographical errors; thorough editing is advised for the revision.
    Response: We have checked the tables.

  5. Larger eQTL dataset, L134-135: The URLs provided should be moved to the reference section.
    Response: We have shifted the websites to the reference section.

  6. Figures 2 and 3: It would be helpful to include the unit for "Density" on the y-axis. Additionally, ensure consistency in the y-axis scales between the two figures; one may represent relative density while the other should display percentage density.
    Response: The y-axis of the density curves does not have units. The area under the density curve has to sum to one. In Figure 2, polygenic risk scores (PRS) ranged from -2.5 to 2.5. While in Figure 3, the measure of discriminatory ability (AUC) ranged from 0.5 to 0.6. To make the area under the density curve equal 1, such that the partial area under the curve has the interpretation of a proportion of values falling in that range (https://proclusacademy.com/blog/area-under-density-curve-percentile/). Hence the y-axis of both figures looks to be of different scales. For clarity, we have added to the figure legends, “The area under the density curve equals one”.

  7. Figure 4: The proportion of the area does not accurately reflect the number of cases; consider adjusting this for improved clarity.
    Response: We have edited the area to reflect the number of cases.

  8. Polygenic risk score (PRS), L389: Consider labeling the equation for clarity and reference.
    Response: We have added the label for equation 1, and referred to it in the following sentence “PRS is estimated as the weighted sum of effect alleles in n single nucleotide poly-morphisms (SNPs) found to be associated with breast cancer (Equation (1)).”  
